# Demolition Activity and Elevated Blood Lead Levels among Children in Detroit, Michigan, 2014–2018

**DOI:** 10.3390/ijerph17176018

**Published:** 2020-08-19

**Authors:** Carla Bezold, Samantha J. Bauer, Jessie P. Buckley, Stuart Batterman, Haifa Haroon, Lauren Fink

**Affiliations:** 1Detroit Health Department, Detroit, MI 48201, USA; bauers@detroitmi.gov (S.J.B.); haroonh@detroitmi.gov (H.H.); finkl@detroitmi.gov (L.F.); 2Department of Environmental Health and Engineering, Johns Hopkins University, Baltimore, MD 21205, USA; jbuckl19@jhu.edu; 3Department of Environmental Health Sciences, School of Public Health, University of Michigan, Ann Arbor, MI 48109, USA; stuartb@umich.edu

**Keywords:** demolitions, lead, children, elevated blood lead level, negative control

## Abstract

Older buildings in the United States often contain lead paint, and their demolition poses the risk of community lead exposure. We investigated associations between demolitions and elevated blood lead levels (EBLLs) among Detroit children aged <6 years, 2014–2018, and evaluated yearly variation given health and safety controls implemented during this time. Case-control analysis included incident EBLL cases (≥5 µg/dL) and non-EBLL controls from test results reported to the Michigan Department of Health and Human Services. Exposure was defined as the number of demolitions (0, 1, 2+) within 400 feet of the child’s residence 45 days before the blood test. We used logistic regression to calculate odds ratios (ORs) and 95% confidence intervals (CIs), and test effect modification by year. Associations between demolition and EBLL differed yearly (*p* = 0.07): 2+ demolitions were associated with increased odds of EBLLs in 2014 (OR: 1.75; 95% CI: (1.17, 2.55), 2016 (2.36; 1.53, 3.55) and 2017 (2.16; 1.24, 3.60), but not in 2018 (0.94; 0.41, 1.86). This pattern remained consistent in sensitivity analyses. The null association in 2018 may be related to increased health and safety controls. Maintenance of controls and monitoring are essential, along with other interventions to minimize lead exposure, especially for susceptible populations.

## 1. Introduction

The presence of vacant or blighted properties poses health concerns for nearby residents including stress, reduced physical activity, and poor mental health [1,2,3]. Vacant or blighted properties may also contribute to certain types of crime [4,5]. Demolition is one strategy to address the public health problem of vacant or blighted properties and can be an important tool for urban renewal and advancement of public health. Alongside the potential benefits of demolition, potential adverse impacts must also be considered. When a home containing lead paint is demolished, lead dust can be entrained into the air, settle on the ground and other surfaces, and potentially expose residents. Lead in dust fall from housing demolitions and related debris removal has been measured in Baltimore [6] and Chicago [7]. A 2007 study found an association between multiple demolitions in a census block and children’s blood lead levels within 45 ensuing days in St. Louis City, Missouri [8].

In Detroit, Michigan, USA, which has the nation’s largest demolition program, demolition is an important part of the city’s revitalization. In 2014, the City’s Blight Reduction Task Force identified more than 40,000 blighted structures, with an additional 39,000 at risk for becoming blighted. As of 2019, the city had completed 19,000 residential demolitions [9]. Emerging evidence points to benefits of this large-scale demolition effort including lower levels of crime [10] and economic benefits [11].

More than 90% of Detroit homes were built prior to 1978, when restrictions on the lead content in paint for residential uses were imposed. Due in part to this aging housing, lead exposure among children in Detroit is higher than in the rest of the state of Michigan. In 2016, 8.8% of Detroit children tested had an elevated blood lead level (EBLL), defined as ≥5 µg/dL, compared to less than 4% in Michigan overall [12]. Given the existing burden of lead exposure in the city and evidence from other jurisdictions regarding demolitions, the City of Detroit has worked to monitor and address the potential public health impacts of demolitions over the last five years.

A preliminary analysis by the Detroit Health Department in 2017 suggested a possible association between demolitions and EBLLs [13]. In light of these preliminary findings, adjustments were made to the demolition program. While some dust mitigation procedures such as wetting were practiced prior to 2017, the protocols were strengthened as a result of this preliminary analysis. The use of hoses to wet down material has been shown to be effective at reducing fugitive dust if the water is applied before and during demolition [14] and during debris removal [6,7]. Detroit’s protocols specify that contractors knock a hole in the roof, pre-wet the structure through the hole for at least five minutes, wet the building’s exterior on all sides, direct water at the point of demolition throughout demolition activity, and spray water during debris removal activities [15]. Lead-containing debris and settled dust (including dust knocked down by the water spray) that remains on the site after sweeping and cleanup will be gradually incorporated into deeper soils and diluted, with generally limited potential for human exposure if the site is covered with clean soils and vegetation. The program now deploys on-site inspectors at more than 90% of sites to ensure enforcement of health and safety protocols. Other adjustments include a partial precautionary moratorium on demolitions in the months of May–September 2018 in the five ZIP codes with the highest proportion of children with EBLLs, prohibiting demolitions on windy days, and providing education and outreach to residents. These policies and activities were implemented across late 2017 and early 2018 and remain in place currently.

The objectives of this study were to describe the spatial and temporal trends in demolitions and to evaluate possible associations with EBLLs among children ≤5 years old during 2014–2018 in Detroit. This evaluation can help to verify that measures undertaken in demolitions are sufficient and health-protective, potentially informing policies in Detroit and elsewhere. 

## 2. Materials and Methods 

### 2.1. Source Population and Data Sources

The source population comprised children aged <6 years who resided in Detroit and had a blood lead test reported to the Michigan Department of Health and Human Services (MDHHS) during 2014–2018. All blood lead tests performed on children less than 6 years old are required to be reported to MDHHS under the Michigan Public Health Code [16]. Tests were reported for Detroit children aged <6 years. Testing is required by Michigan’s Medicaid program for children at ages 12 and 24 months, though only 62% of Medicaid-enrolled children in Detroit were tested by their second birthday, and only 74% were tested before their third birthday [17]. The dataset includes all tests conducted from January 2012 through December 2018 and reported through May 2019 (N = 211,733), as test reports are sometimes delayed. Tests that did not have a valid Detroit address (N = 34,004) or were for individuals ≥6 years old (N = 2544) were removed from the dataset. 

#### 2.1.1. Definitions of Incident and Prevalent Elevated Blood Lead Levels (EBLLs)

Individuals were considered cases of EBLL if they had a blood lead test with a value ≥5 µg/dL, based on results rounded to the nearest whole number. Prevalent cases were identified for each year consistent with MDHHS guidance: If a child had more than one blood lead test reported in a given calendar year, we selected the highest venous result where available. If no venous test was available, we retained the highest result from a capillary test. If the only available test was unknown sample type, we retained that value [12]. Incident tests were defined as the first venous test ≥5 µg/dL during the study period, or the first capillary test ≥5 µg/dL if no venous test was available. We limited our analysis to the period 2014–2018 because the volume of demolitions in Detroit was scaled up significantly at that time, but results from 2012 and 2013 were used to help to distinguish incident and prevalent results in later years. Blood lead test results are considered public health surveillance and informed consent was not required.

Demolition dates and locations are published regularly by the City of Detroit’s demolition program on the City of Detroit’s Open Data Portal. A complete list of all demolitions to date was downloaded in March 2019. Demolitions were geocoded and assigned to census tracts using the City of Detroit’s geocoder. Census tract median housing age was determined using the American Community Survey (ACS) 5 year average (2013–2017) table B25035 and joined to demolitions based on the census tract number. All analyses were conducted using the RStudio version 3.5.2 (RStudio: Boston, MA, United States) (2018) “Eggshell Igloo” statistical computing platform and Arc GIS Pro 2.4.2 (Esri Inc.: Redlands, CA, USA) (2019), geostatistical software.

#### 2.1.2. Descriptive Analysis

We described patterns in demolitions and EBLLs in the source population in several ways. We mapped the total number of demolitions in each census tract over the study period using ArcGIS Pro. Next, we calculated the percent EBLL for each year during the study period and tested for variation across time using a chi-square test. We also calculated the percent EBLL by ZIP Code, which is the geographic unit that has been used to prioritize areas for lead prevention activities, including the precautionary moratorium on demolitions imposed in 2018. We also mapped incident EBLLs over the study period using a kernel density method with a radius of 10. This method is used to calculate the density of point features around an output raster cell and illustrates areas of the city with the highest density of incident EBLLs. 

### 2.2. Case-Control Analysis

To evaluate potential associations between demolitions and incident EBLLs, we conducted a retrospective case-control study. Cases were incident EBLLs, defined as above, and controls were individuals without known EBLL during the study period. For controls, the collection date of the first venous test was used as the incident test date unless no venous test was available, in which case the collection date of the first capillary or unknown test type was used.

#### Exposure—Proximate Demolition Activity

For each incident test result, we summed the number of demolitions that occurred within a 400 foot radius circular buffer around the child’s home in the 45 days prior to the incident test date. The 400 foot exposure buffer was based on evidence suggesting this was the maximum distance likely to experience dustfall from a demolition [7]. The 45-day period corresponds to the half-life of lead in blood [18]. Demolition exposure was categorized as 0, 1, or 2+ demolitions during each time window for the primary analysis. We also calculated a negative control exposure variable, defined as the number of demolitions within a 400 foot radius circular buffer that occurred 45 days after the incident test date. The negative control exposure approach is used to evaluate potential unmeasured confounding and has been described in detail elsewhere [19,20,21]. Our analysis assumes that demolitions occurring after the incident test date are not causally related to the test result and that the main exposure and negative control exposure variables share any unmeasured confounders of the EBLL relationship. Because we assume no causal association of the negative control exposure with EBLL, any observed association between the negative control exposure and EBLLs (after adjusting for the main exposure) would suggest that both the negative control and main exposure associations with EBLL are confounded and not attributable to demolitions.

### 2.3. Statistical Analysis

Odds ratios (ORs) and 95% confidence intervals (CIs) were calculated using logistic regression models adjusted for the matching factors: age category (<1, 1–2, 3–5 years), gender, specimen type (venous, capillary, unknown), season (summer versus not), calendar year, and ZIP code. Data were analyzed for all years together and stratified by year. We tested for effect measure modification, to test whether the association between demolitions and EBLLs differed by year, using a likelihood ratio test comparing a fully adjusted model that included an interaction term between demolition exposure and year (coded as a categorical variable) to one without. Our primary analysis included both the main exposure variable (demolitions occurring 45 days before a test) and the negative control exposure variable (demolitions occurring 45 days after a test). We considered results to be statistically significant at *p* < 0.05 for main effects and *p* < 0.1 for the likelihood ratio tests assessing effect measure modification.

### 2.4. Sensitivity Analysis

To assess the potential impact of modeling decisions on our findings, we conducted several sensitivity analyses. To evaluate the influence of including the negative control exposure, we estimated associations in models that included the main exposure variable only. We assessed alternate exposure time windows (15, 30, and 60 days) to compare with our main exposure based on a 45 day window. We repeated the analysis using a 200 foot buffer to assess whether associations were stronger when demolitions were closer. We also repeated our primary analysis using only venous measurements because blood lead tests from a capillary sample may be prone to misclassification due to their relatively high false positive rate [22].

Our primary analysis employs a case-control design with the goal of increasing validity, but this limits sample size and precision. To test our results in a larger sample size, we repeated the primary analysis (demolitions within 400 feet in the 45 days prior to a test, stratified by year) in a cross-sectional study of the full source population that included every child with a test available. We estimated prevalence odds ratios (PORs) and 95% CIs in logistic regression models adjusting for the matching factors used in the case-control analysis.

The five ZIP codes with consistently high percentages of EBLL among children tested were the focus of the 2018 interventions, including the summertime moratorium. We evaluated potential associations between demolitions and EBLL in these five ZIP codes separately and with the rest of the city.

## 3. Results

### 3.1. Descriptive Analysis

City-wide, the prevalence of EBLL varied significantly across years (*p* < 0.01), from 8.8% in 2016 to 7.1% in 2018 (Table 1). The number of children tested also varied, from a high of 23,408 in 2016 to a low of 19,820 in 2018. These numbers represent one observation per child per year. If a child had multiple tests in a year, only the highest result from a venous sample was retained. Figure 1 shows a heat map of incident EBLLs for the study period overall. The largest density of new cases were in the southwest and west parts of the city. This is likely due in part to the relatively higher number of households with young children in these areas. In contrast, Figure 2 shows the percentage of EBLLs among children tested in 2016 by ZIP Code. ZIP Codes 48202, 48204, 48206, 48213, and 48214, outlined in white in the figure, consistently have among the highest percentages of EBLLs in the city. In 2016, EBLL in these five ZIP Codes ranged from 13.6% in 48213 to 22.2% in 48206. Based on 2016 numbers, these five ZIP Codes were the focus of prevention activities and the 2018 demolition moratorium.

Figure 3 shows the distribution of demolitions by census tract over the study period. Table 2 shows the distribution of demolition exposure by blood lead levels during the study period. The majority (95%) of children were not exposed to any demolitions within 400 feet of their residence in the 45 days prior to their blood lead test. Over the 5-year study period, 87 children lived within 400 feet of 5 demolitions in the 45 days prior to a blood lead test, and 19 were exposed to more than 10 in that time period.

### 3.2. Case-Control Analysis

Table 3 presents descriptive characteristics of cases and controls. Cases were slightly more likely to be male (54%) than female, and 61% of cases were ages 1–2 years old. Nearly 80% of cases were identified using venous samples, although the proportion of tests based on venous samples varied slightly across the years from 86% in 2014 to 72% in 2018 (84% in 2015, 77% in 2016, 75% in 2017).

Table 4 presents results for the associations between demolitions 45 days before (main exposure) and after (negative control exposure) a blood test and incident EBLL, overall and stratified by year. The association between demolitions before the test and EBLL varied across years (*p* = 0.07). There was suggestive evidence of an association between demolitions 45 days before a test and EBLL in some years, particularly 2016 (OR for 2+ demolitions: 2.36, 95% CI: 1.53, 3.55) and 2017 (OR for 2+ demolitions = 2.16, 95% CI 1.24, 3.60). In contrast, there was no evidence of this association in 2018 (0.94, 95% CI: 0.41, 1.86). In addition, we found no statistically significant associations between the negative control exposure and EBLL, suggesting no strong unmeasured confounding of demolition and EBLL associations.

### 3.3. Sensitivity Analyses

The trend across years was similar in the five target ZIP Codes and the rest of the city (Appendix A). The lack of an association in 2018 in ZIP Codes that were not the target of the moratorium is notable. The overall pattern of association between demolitions and EBLL across years was similar using 15-, 30-, and 60-day exposure windows, although associations were slightly stronger using a 15- or 30-day time window relative to a 45-day window (Appendix A). There were no significant associations between demolition exposure and EBLL in 2018 using any time window. When the exposure buffer was reduced to 200 feet, the association between demolitions and EBLL remained significantly elevated in 2016 (Appendix A), but there was considerable variability in the estimates, likely due to the small number of cases exposed to multiple demolitions at such close proximity. Findings were similar when the case definition was limited to incident venous EBLL (Appendix A). Our cross-sectional analysis of all available blood lead testing data in the source population produced similar results to the case-control analysis (Appendix A). Results of a binary exposure variable, any demolitions versus no demolitions, also showed a similar pattern (Appendix A).

## 4. Discussion

We did not observe associations between demolitions and EBLLs in 2018, contrary to indications of an association in many earlier years. Of note, the lack of an association in 2018 was observed across the entire city and persisted when the affected ZIP Codes were removed from the model, suggesting it may be independent of the summer restriction on demolitions in some areas.

The variation across years may result from dust controls and other management practices implemented in Detroit’s large-scale program being successful. In addition, this variation may be attributable to differences in the types of homes that were demolished across the years. Demolitions in 2018 occurred in areas with slightly newer housing stock relative to 2016 demolitions. However, nearly all demolitions during the study period occurred in census tracts where the median home construction year precedes 1960. Thus, differences in the types or ages of homes demolished alone are unlikely to explain the year-to-year variation.

The burden of elevated blood lead in Detroit remains significant. In the state of Michigan, fewer than 4% of children tested were found to have elevated blood lead levels in 2016. During the study period, prevalence of EBLL in Detroit varied between 7.1% and 8.8%, approximately twice the statewide prevalence. The age of Detroit’s housing stock, along with declining quality due to poor maintenance, suggest that peeling paint and dust cause the majority of childhood lead exposure in Detroit. This is consistent with the primary source of exposure identified in other urban areas [22,23]. Many Detroit homes utilize lead service lines, and thus, drinking water is another potential source of exposure, although state and federal guidelines for lead are attained. Lead exposure prevention should target primary prevention through increased access to and funding for abatement and critical home repairs, enforcing landlord requirements for lead-safe rental housing, and education about lead sources, cleaning, and lead exposure avoidance.

### Limitations and Strengths

As with any epidemiologic study, residual confounding could affect results, for example, if neighborhoods with a high number of demolitions also have a high number of other lead exposures. We employed a negative control exposure approach to quantify the impact of potential unmeasured confounding by neighborhood factors that remain constant during the 45 days immediately before and after a demolition. The lack of a statistically significant association between the negative control exposure variables and EBLL in any of the study years suggests that results of our main analysis are not strongly confounded by time-invariant neighborhood factors, increasing the robustness of our findings.

Our findings depend on the choice of buffer size and exposure time window. The association between demolitions and EBLL, in the years where one was observed, was stronger for exposures closer in time (15 versus 45 days) and space (200 versus 400 feet). A number of these sensitivity analyses were limited by small numbers of exposed cases and underpowered to detect an effect should one exist. The strong associations observed in some years despite this lack of power suggest the potential for demolitions to expose nearby children in the absence of proper controls, and the need to ensure controls to reduce exposure.

This analysis relied on data collected for public health surveillance. In order to be included as either a case or a control, an individual had to have a blood lead test reported to the state health department. The public health recommendation for lead testing in Detroit is that all children be tested at least twice before age 3 as part of routine pediatric care, and between ages 3 and 5 if not previously tested. While testing is required by Michigan’s Medicaid program for children at ages 12 and 24 months, only 62% of children in Detroit enrolled in Medicaid were tested at least once before their second birthday and only 74% before their third birthday. About 30% of children in our study population had their incident test between ages 3 and 5, consistent with the fact that universal testing takes place early in life. In addition to the recommendations for routine testing, there may be testing performed in response to a suspected exposure. If testing patterns changed across years, the testing patterns could introduce variability into the association between demolitions and EBLL not related to the demolition program. We also recognize limitations due to the use of capillary blood lead measurements, including false positives and higher detection limits; more sensitive and precise measurements would facilitate analyses using blood lead levels as continuous exposure measure.

We hypothesize the changes in the association over time might reflect changes in demolition protocols, a selection of which are outlined in Appendix B. Data are available that document implementation of some of these changes. For example, DHD has access to checklists completed by on-site field inspectors verifying that safety protocols are being followed. However, we are unable to verify whether all improved safety precautions, particularly those related to notification, were consistently implemented. The interaction between demolition exposure and calendar year was suggestive, and we cannot rule out chance as an explanation for the differences. In addition, meteorological variability in rainfall, wind patterns, and temperature may have contributed to differences in associations by year. Potentially, exposure misclassification could be reduced by incorporating information on wind direction and other meteorological factors affecting dispersion of lead-containing fugitive dust, along with a determination whether children were downwind during the demolition. However, an exposure metric based on proximity to a demolition event is a simple and potentially robust indicator of exposure, particularly considering the potential errors that may be introduced by indicators that include wind direction and other information given the variability of local wind fields and the emissions and dispersion of fugitive dusts.

Demolitions in Detroit are currently clustered in neighborhoods to maximize efficiency and impact. As such, exposure is not randomly distributed across the city. The temporal variation seen in associations between demolitions and EBLL could be due to differences in the underlying housing stock being demolished. For example, demolitions in 2018 occurred less frequently in areas where the median home construction year is prior to 1940 (Appendix A). However, nearly all demolitions occurred in neighborhoods built prior to 1960, so we expect some amount of lead dustfall associated with any demolition in Detroit.

The study has important strengths. The size and scope of Detroit’s demolition program allows for a larger sample of individuals exposed to demolitions than many previous studies. Previous studies have measured lead in dust fall from demolitions [6,7]; our analysis is among the first to directly investigate a potential association between demolitions and children’s blood lead levels. One previous study of which we are aware observed an association between one or more demolitions at the census block level and children’s blood lead levels [8]. Our study builds on previous findings by employing individual-level estimates of both the exposure and the outcome. Our sensitivity analyses demonstrate that our findings were robust to the choice of buffer size, exposure time window, and analytical approach. The use of a negative control exposure, an emerging tool in environmental epidemiology, further strengthens our findings by demonstrating no strong unmeasured confounding of our associations.

## 5. Conclusions

Demolition of blighted houses can have important consequences for public health and safety [10]. Just over 5% of children were exposed to demolitions within 45 days of a blood lead test, but the volume of demolitions occurring across the city suggests many more children are exposed to one or more demolitions at some point during childhood. Given the scale and speed with which demolitions have been conducted in Detroit, it is essential to continue to monitor for unintended consequences of these efforts and ensure that vulnerable populations are protected from lead exposure and other hazards. There was no evidence of an association between demolitions and EBLLs in 2018. This finding does not appear to be associated with the moratorium on demolitions in select ZIP codes, suggesting that demolitions can be done throughout the city in a manner that is protective of health. The remainder of the health and safety protocols currently in place should remain for future demolitions and can serve as a model for residential demolitions nationwide.

While this study suggests that demolitions can be performed without the risk of significant lead exposure to children in Detroit, lead exposure remains a serious environmental health and environmental justice concern. As noted, deteriorated paint and overall housing quality remain problematic in many portions of the city, and poverty and other risk factors increase the vulnerability and susceptibility of children to elevated blood lead levels. Public health efforts should focus on strategies to reduce exposure to lead, including increasing the supply of lead-safe housing through abatement and enforcement of existing ordinances and education regarding lead exposure avoidance.

## Figures and Tables

**Figure 1 ijerph-17-06018-f001:**
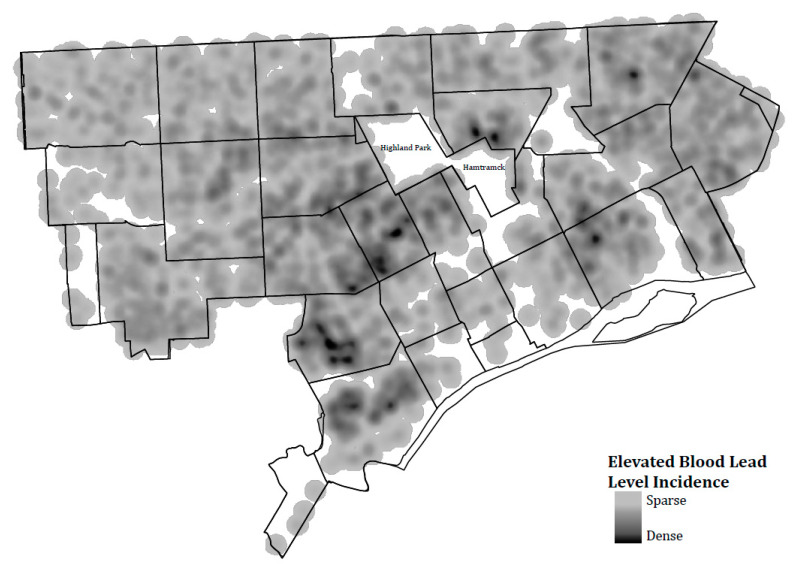
Incident cases of elevated blood lead levels among children under 6, Detroit, 2014–2018. Note: Heat map based on kernel density method, which calculates the density of point features around each output raster cell, with a feature count radius of 10. Elevated blood lead level incidence includes both venous and capillary lead tests. Data from the Michigan Department of Health and Human Services Data Warehouse.

**Figure 2 ijerph-17-06018-f002:**
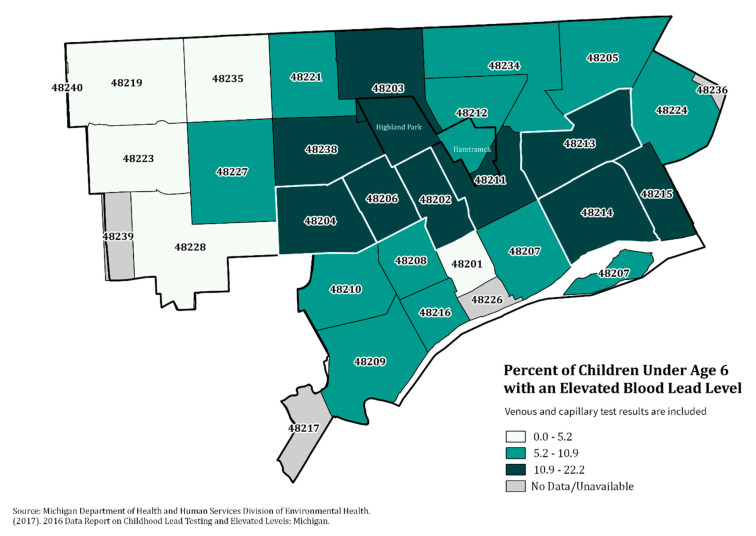
Percentage of children tested identified to have elevated blood lead levels by ZIP code, Detroit, 2016. Note: Data from the Michigan Department of Health and Human Services Division of Environmental Health (version 6/16/2018), 2016 Data Report on Childhood Lead Testing and Elevated Levels: Michigan. Percentage of children with elevated blood lead levels includes both venous and capillary tests. Zip Codes outlined in white indicate the 5 areas where the demolition moratorium was implemented in 2018.

**Figure 3 ijerph-17-06018-f003:**
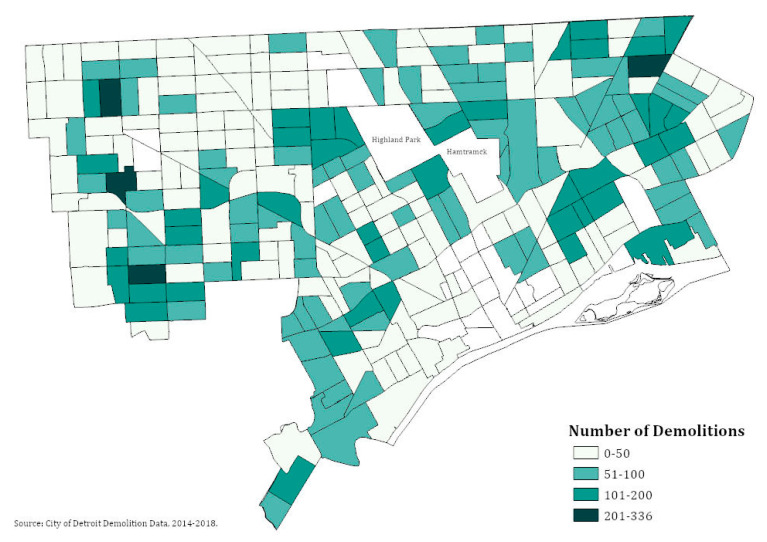
Number of demolitions by census tract in Detroit, 2014–2018. Note: Demolition data from City of Detroit Open Data Portal. Numbers represent the total number of demolitions between January 2014 and December 2018.

**Table 1 ijerph-17-06018-t001:** Prevalence of elevated blood lead levels in Detroit, 2014–2018.

	2018	2017	2016	2015	2014
Number of children <6 years old with a reported blood lead test (one observation per child per year)	19,820	22,216	23,408	21,404	23,134
EBLL (≥5 µg/dL)	1406	1658	2055	1624	1915
Percent EBLL (%)	7.1	7.5	8.8	7.6	8.3

Note: Elevated blood lead level (EBLL). Data from the Michigan Department of Health and Human Services Data Warehouse. The table includes one observation per child per year. Children can be included in multiple years. If a child had multiple tests in a year, only the highest result from a venous sample was retained. If no result from a venous sample was available, the highest result from a capillary or unknown sample was used.

**Table 2 ijerph-17-06018-t002:** Demolition exposure by children’s blood lead level, Detroit, 2014–2018 (N = 109,982 observations).

			Number of Demolitions within 400 Feet of Residence in 45 Days Prior to Blood Lead Test										
Year		Total	0	1	2	3	4	5	6	7	8	9	≥10
2014	BLL < 5 µg/dL	21,219	20,075	787	193	73	32	26	11	5	5	5	7
	BLL ≥5 µg/dL	1915	1768	90	27	14	8	*	*	0	*	0	*
2015	BLL < 5 µg/dL	19,780	18,561	867	200	80	36	22	*	7	*	0	*
	BLL ≥ 5 µg/dL	1624	1505	93	14	*	9	0	0	0	0	0	0
2016	BLL < 5 µg/dL	21,353	20,217	906	168	32	17	*	*	*	*	*	*
	BLL ≥ 5 µg/dL	2055	1901	106	29	7	5	*	*	0	0	0	*
2017	BLL < 5 µg/dL	20,558	19,851	580	78	21	14	*	6	*	*	0	*
	BLL ≥ 5 µg/dL	1658	1558	72	13	8	*	0	*	0	0	0	*
2018	BLL < 5 µg/dL	18,414	17,516	717	116	40	18	*	*	0	0	0	*
	BLL ≥ 5 µg/dL	1406	1342	48	13	*	0	0	*	0	0	0	0
2014–2018	BLL < 5 µg/dL	101,324	96,220	3857	755	246	117	59	24	16	9	8	13
	BLL ≥ 5 µg/dL	8658	8074	409	96	34	24	*	10	0	*	0	6

Note: Blood lead level (BLL). Data from the Michigan Department of Health and Human Services Data Warehouse. The table includes one observation per child per year. Children can be included in multiple years. If a child had multiple tests in a year, only the highest result from a venous sample was retained. If no result from a venous sample was available, the highest result from a capillary or unknown sample was used. * indicates data suppressed due to a cell count <5.

**Table 3 ijerph-17-06018-t003:** Descriptive characteristics of cases (incident elevated blood lead levels among children <6 years old) and controls (N = 54,150 observations, 5430 cases), Detroit, 2014–2018.

		Cases N = 5430	Controls N = 48,720
		*n* (%)	*n* (%)
Age	<1 year	279(5)	7613(16)
	1–2 years	3335(61)	26,862(55)
	3–5 years	1816(33)	14,245(29)
Gender	F	2416(44)	23,967(49)
	M	2956(54)	24,276(50)
	Unknown	58(1)	477(1)
Month	January	307(6)	3836(8)
	February	250(5)	3579(7)
	March	296(5)	4239(9)
	April	323(6)	3977(8)
	May	398(7)	4108(8)
	June	568(10)	3875(8)
	July	578(11)	3826(8)
	August	730(13)	4973(10)
	September	813(15)	5464(11)
	October	537(10)	4381(9)
	November	370(7)	3495(7)
	December	260(5)	2967(6)
Year	2014	1206(22)	9972(20)
	2015	995(18)	8835(18)
	2016	1269(23)	10,169(21)
	2017	1013(19)	10,101(21)
	2018	947(17)	9643(20)
Specimen Type	Capillary	1129(21)	10,389(21)
	Venous	4300(79)	38,011(78)
	Unknown	1(0)	320(1)
Demolitions within 400 feet, 45 days before test	0	5054(93)	46,419(95)
	1	255(5)	1755(4)
	2	121(2)	546(1)

Note: Cases were defined as the first venous test with a result ≥5 µg/dL; for individuals who did not have any venous tests during 2012–2018, the first test from a capillary or unknown specimen type with a result ≥5 µg/dL was considered an incident EBLL. The date the specimen was collected was used as the incident test date. Controls were individuals without an EBLL during the study period; the incident test date was the first result from a venous test where available and from a capillary test where no venous test was available.

**Table 4 ijerph-17-06018-t004:** Results of a case-control analysis of the association between demolitions and incident elevated blood lead levels among children <6 years old, by calendar year, Detroit, 2014–2018 (N = 54,150 observations, 5430 cases).

	2018	2017	2016	2015	2014	Overall
Incident EBLL cases	947	1013	1269	995	1206	5430
0 Demolitions before test	900	950	1171	918	1115	5054
1 Demolition before test	39	43	63	59	51	255
2 Demolitions before test	8	20	35	18	40	121
Controls	9643	10,101	10,169	8835	9972	48,720
0 Demolitions before test	9197	9762	9662	8312	9486	46,419
1 Demolition before test	356	270	414	373	342	1755
2 Demolitions before test	90	69	93	150	144	546
Main analysis (includes demolitions 45 days before and after test)	OR 95% CI	OR 95% CI	OR 95% CI	OR 95% CI	OR 95% CI	OR 95% CI
1 Demolition before test (main exposure)	0.96	1.51	1.24	1.32	1.03	1.19
	(0.67, 1.33)	(1.06, 2.10)	(0.92, 1.63)	(0.98, 1.76)	(0.74, 1.39)	(1.03, 1.36)
2 or More Demolitions before test (main exposure)	0.94	2.16	2.36	1.03	1.75	1.63
	(0.41, 1.86)	(1.24, 3.60)	(1.53, 3.55)	(0.60, 1.68)	(1.17, 2.55)	(1.32, 2.00)
1 Demolition after test (negative control exposure)	0.96	1.36	1.17	1.01	1.28	1.14
	(0.67, 1.34)	(0.96, 1.87)	(0.84, 1.61)	(0.72, 1.37)	(0.96, 1.69)	(0.99, 1.31)
2 or More Demolitions after test (negative control exposure)	1.00	1.46	0.86	0.89	1.39	1.11
	(0.53, 1.73)	(0.81, 2.49)	(0.46, 1.49)	(0.47, 1.55)	(0.89, 2.11)	(0.88, 1.40)
Demolitions before test only	OR 95% CI	OR 95% CI	OR 95% CI	OR 95% CI	OR 95% CI	OR 95% CI
1 Demolition	0.95	1.52	1.24	1.32	1.05	1.19
	(0.67, 1.33)	(1.07, 2.11)	(0.92, 1.63)	(0.98, 1.76)	(0.76, 1.42)	(1.04, 1.37)
2 or More Demolitions	0.94	2.30	2.35	1.03	1.89	1.66
	(0.41, 1.86)	(1.33, 3.82)	(1.53, 3.55)	(0.60, 1.67)	(1.28, 2.73)	(1.35, 2.04)

Note: Cases were defined as the first venous test with a result ≥5 µg/dL; for individuals who did not have any venous tests during 2012–2018, the first test from a capillary or unknown specimen type with a result ≥5 µg/dL was considered an incident EBLL. The date the specimen was collected was used as the incident test date. Controls were individuals without an EBLL during the study period; the incident test date was the first result from a venous where available and from a capillary test where no venous was available. Results were estimated in logistic regression models adjusted for specimen type (venous, capillary, unknown), month and year of collection, age category (<1, 1–2, 3–5 years), gender, and ZIP code.

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
