# Peer review of "Demolition Activity and Elevated Blood Lead Levels among Children in Detroit, Michigan, 2014–2018"

_ijerph, 2020, doi:10.3390/ijerph17176018_

Round 1

Reviewer 1 Report

This manuscript make a good case for taking practical measures to prevent child lead exposure from inner city demolition of old houses.  Several minor suggestions are below.

[1] Lines 50-62 describe the wetting procedures to limit airborne lead dispersal during demolition.  The reader would benefit from information on the fate of the lead in the remaining debris and whether some lead becomes dispersed by suspension in the water used for wetting the structures.

[2] For those lacking expertise in epidemiology, the importance of the phrase “p<0.1 for effect measure modification” (lines 133-134) would be helpful.

[3] The importance and relevance of Figure 1 and the “kernel density method” (line 175) are not explained.

[4] The various tables stratify the OR by 1 vs 2 or more demolitions.  This makes sense, but the reader would like to see the values for the combination (1 or more demolitions) for comparison.

Author Response

Response to Reviewer #1: 

This manuscript make a good case for taking practical measures to prevent child lead exposure from inner city demolition of old houses. Several minor suggestions are below.

We appreciate you seeing the value in our research and helping us to clarify certain points so that it may be of greater use to its readers.

Lines 50-62 describe the wetting procedures to limit airborne lead dispersal during demolition.  The reader would benefit from information on the fate of the lead in the remaining debris and whether some lead becomes dispersed by suspension in the water used for wetting the structures.

Thank you for your suggestion.  While we did not study the fate of lead either in the debris or water used to control dust, most of this lead is in particulate form and will become incorporated into local surface soils, as evidenced by elevated lead levels in many urban soils.  Over time, the surface soils will become more mixed, and some downward migration of lead will occur, both with the effect of decreasing surface concentrations that have the potential for exposure.  In the text we note that: “Lead-containing debris and settled dust (including dust knocked down by the water spray) that remains on the site after sweeping and cleanup will be gradually incorporated into deeper soils and diluted, with generally limited potential for human exposure if the site is covered with clean soils and vegetation.”

or those lacking expertise in epidemiology, the importance of the phrase “p<0.1 for ffect measure modification” (lines 133-134) would be helpful.

We appreciate the suggestion to clarify this point for diverse readers. We clarified the definition of effect measure modification in lines 136-138 and have added a clarification about the choice of p-value in lines 141-142.

The importance and relevance of Figure 1 and the “kernel density method” (line 175) are not explained.

Thank you for identifying this gap. We have added a sentence to the methods (109-112) addressing this and have also added language to the note for Figure 1.

The various tables stratify the OR by 1 vs 2 or more demolitions.  This makes sense, but the reader would like to see the values for the combination (1 or more demolitions) for comparison.

Thank you for this suggestion. We have addressed this in supplementary Table 6S.

Reviewer 2 Report

This is a well-written and interesting manuscript on the impact of housing demolition on children’s blood lead level exceedances. It makes use of a large dataset and applies a number of valuable methods to evaluate potential confounding and robustness of the findings.

Comment 1: The methods lack information on the blood lead test regime that applies to this study population.

The methods do not make clear what the criteria are for performing a lead blood test in this population. Do all children at some point receive a blood test? What could trigger such a test? Will a test be performed if there is suspected exposure to lead? It is not clear how representative the blood lead levels are for the source population, or for what % of the population of <5 year olds a test is available.

Only in the discussion it is mentioned that testing is required by Michigan’s Medicaid program for children at ages 12 and 24 months. This type of information should be moved to the methods, to provide the reader with a better understanding of blood lead tests in this population. Why are 30% of the tests in 3-5 year olds if Medicaid funds tests in 12/24 months of age? Are these delayed tests that should have happened at age 12/24 months?

Comment 2: Table 1: these represent the number of tests, not the number of children, correct? Needs to be clearly stated.

Comment 3: Comparing numbers in table 1 with table 3: Is it correct there are on average about 2 tests per child?

Comment 4: Table 4: what are the numbers of exposed cases and controls? This is listed in Table 3 but would be good to include in Table 4 as well, for each year.

Comment 5: The discussion mentions that “The age of Detroit’s housing stock, along with declining quality due to poor maintenance, suggest that peeling paint and dust cause the majority of childhood lead exposure in Detroit.” I do not consider this a very convincing argument. Do the authors have access to other information or references that indicate that peeling paint and paint dust is the main source of lead exposure for children living in Detroit?

Author Response

Response to Reviewer #2: 

This is a well-written and interesting manuscript on the impact of housing demolition on children’s blood lead level exceedances. It makes use of a large dataset and applies a number of valuable methods to evaluate potential confounding and robustness of the findings.

We appreciate you seeing the value in our research and pushing us further to better describe the methods.

Comment 1: The methods lack information on the blood lead test regime that applies to this study population.

The methods do not make clear what the criteria are for performing a lead blood test in this population. Do all children at some point receive a blood test? What could trigger such a test? Will a test be performed if there is suspected exposure to lead? It is not clear how representative the blood lead levels are for the source population, or for what % of the population of <5 year olds a test is available.

Only in the discussion it is mentioned that testing is required by Michigan’s Medicaid program for children at ages 12 and 24 months. This type of information should be moved to the methods, to provide the reader with a better understanding of blood lead tests in this population. Why are 30% of the tests in 3-5 year olds if Medicaid funds tests in 12/24 months of age? Are these delayed tests that should have happened at age 12/24 months?

Thank you for this important and detailed comment. We have added the information about Medicaid testing and other reasons that children may be tested to the Methods section in lines 73-78.  We have also specified that testing is recommended between ages 3 and 5 for children not tested at age 12 and 24 months, since children are not tested in a timely manner this catch-up testing is performed. We have also added language to the discussion about possible reasons for testing beyond what is recommended as part of routine pediatric practice.

Comment 2: Table 1: these represent the number of tests, not the number of children, correct? Needs to be clearly stated.

Thank you for this suggestion. The numbers represent one observation per child per year. We have made this clearer in Table 1 and added more information in lines 171-172. 

Comment 3: Comparing numbers in table 1 with table 3: Is it correct there are on average about 2 tests per child?

Thank you for this question. The numbers in Table 1 represent one observation per child per year, while the numbers in Table 3 represent individual children defined as either a case with an incident EBLL (first venous test with result ≥5 µg/dL or if no venous test available, first test from a capillary or unknown specimen with a result ≥5 µg/dL). Individuals can be included in multiple years in Table 1, but are counted only once in Table 3.

Comment 4: Table 4: what are the numbers of exposed cases and controls? This is listed in Table 3 but would be good to include in Table 4 as well, for each year.

Thank you for this suggestion. We have added a table (Table 2) showing the detail regarding the distribution of exposure among cases and controls by year and we have added the number of exposed cases and controls for each year to Table 4.

Comment 5: The discussion mentions that “The age of Detroit’s housing stock, along with declining quality due to poor maintenance, suggest that peeling paint and dust cause the majority of childhood lead exposure in Detroit.” I do not consider this a very convincing argument. Do the authors have access to other information or references that indicate that peeling paint and paint dust is the main source of lead exposure for children living in Detroit?

Thank you for this suggestion. We have added a reference to an article about sources of lead exposure in US children that highlights the substantial role paint and dust play in exposure (up to 70% of all exposures) (lines 278-279). In Detroit, more than 90% of housing stock was built prior to 1978 when lead paint was banned.

Reviewer 3 Report

General comments:

Can the authors incorporate historic wind patterns to account for likely direction of dust dispersion during demolition? This may avoid exposure misclassification and strengthen the association between demolition and elevated BLL.

The authors indicate that no dust mitigation practices were used prior to 2018, is this supported by any research or demolition records? Is there any survey information from contractors regarding dust mitigation practices used during demolition prior to the implementation of the law?

Did the authors control for children living in the same household in the logistic regression models?

Are there any unmeasured meteorological factors that could account for the reduction in BLL observed in 2018? Was 2018 hotter and wetter than previous study years? This may cause children to be indoors more in the warmer months and likely reduce their exposure.

Specific comments:

Line 163- can the author provide the ratio of capillary to venous blood test for each of the study years?

Line 185- “Table 3 shows the distribution of demolition exposure by blood lead levels during the study period.” This sentence should be revised to better reflect the data contained in the table.

Author Response

Response to Reviewer #3: 

Can the authors incorporate historic wind patterns to account for likely direction of dust dispersion during demolition? This may avoid exposure misclassification and strengthen the association between demolition and elevated BLL.

Incorporating wind direction and other factors that affect pollutant dispersion could be useful if the site of exposure and the exposure pathways are known. If direct inhalation of fugitive lead-containing dust is responsible, then we need to know if the child is downwind during the short exposure window. If exposure results from both settled and reentrained dust, then we need wind direction information during both demolition and reentrainment periods. Ideally, we would account for effects of trees, large buildings, traffic and other factors that may alter winds and affect dispersion.  We note that Detroit did not allow demolitions with higher wind speeds, where directional effects would be particularly important, and that wind roses for the non-winter months in Detroit show a variety of directions, thus our “naïve” assumption of radial dispersion may not be harmful. However, we now suggest in the Discussion that including wind direction information into the model might help reduce exposure misclassification (Line 320).

An exposure metric based on proximity to a demolition event is a simple and potentially robust indicator of exposure, particularly considering the potential errors that may be introduced by indicators that include wind direction and other information given the variability of local wind fields and the emissions and dispersion of fugitive dusts. Such analyses may be worth attempting and potentially might increase effect sizes although a countering effect is the decrease in the number of children in the exposed group. Overall, we believe that our sample size and approach is sufficient to detect impact of demolitions on children's BLLs

The authors indicate that no dust mitigation practices were used prior to 2018, is this supported by any research or demolition records? Is there any survey information from contractors regarding dust mitigation practices used during demolition prior to the implementation of the law?

Thank you for these questions, which have helped us increase the clarity of the history we present. Appendix A provides a short timeline of wetting and other dust mitigation practices. We have added text to lines 53-54 to clarify that some dust mitigation procedures were practiced prior to 2017, but the protocols were subsequently strengthened.

Did the authors control for children living in the same household in the logistic regression models?

Thank you for this question.

Are there any unmeasured meteorological factors that could account for the reduction in BLL observed in 2018? Was 2018 hotter and wetter than previous study years? This may cause children to be indoors more in the warmer months and likely reduce their exposure.

Thank you for this question and thoughtful suggestion. There was more precipitation in 2018 (43.81 inches) than in 2016 (34.74 inches), but it was cooler (average 51.4 degrees in 2018 versus 52.8 degrees in 2016). Information pertaining to meteorological variables affecting the dispersion of lead specifically when demolitions are occurring might be more relevant than annual data, but these data are not easy to incorporate into the analysis. Additionally, factors that result in more time indoors might actually be expected to increase overall blood lead levels as children spend time in houses where they are exposed to lead hazards. However, the overall prevalence of EBLL in 2018 was lower than observed in 2016. We have added a sentence in the discussion (Line 321) acknowledging the potential role of meteorological factors but do not expect that factors are a primary contributor to the variation in yearly estimates.

Line 163- can the author provide the ratio of capillary to venous blood test for each of the study years?

Thank you for this suggestion. This information has been added beginning in line 218. The proportion of tests that were venous varied from 72% in 2018 to 86% in 2014.

Line 185- “Table 3 shows the distribution of demolition exposure by blood lead levels during the study period.” This sentence should be revised to better reflect the data contained in the table.

Thank you for bringing this to our attention. This was a typo and should refer to Table 2 which was missing from the previous version.